# Biostimulants Improve Plant Growth and Bioactive Compounds of Young Olive Trees under Abiotic Stress Conditions

Giulia Graziani [1,†] , Aurora Cirillo [2,*,†] , Paola Giannini [1] , Stefano Conti [2] , Christophe El-Nakhel [2] , Youssef Rouphael [2] , Alberto Ritieni [1,3] and Claudio Di Vaio [2]

1 Department of Pharmacy, University of Naples Federico II, Via Domenico Montesano 49, 80131 Naples, Italy; giulia.graziani@unina.it (G.G.); paola.giannini@unina.it (P.G.); alberto.ritieni@unina.it (A.R.)
2 Department of Agricultural Sciences, University of Naples Federico II, Via Università 100, 80055 Portici, Italy; stefano.conti@unina.it (S.C.); christophe.elnakhel@unina.it (C.E.-N.); youssef.rouphael@unina.it (Y.R.); claudio.divaio@unina.it (C.D.V.)
3 Unesco Health Education and Sustainable Development, 80131 Naples, Italy
* Correspondence: aurora.cirillo@unina.it
† These authors contributed equally to this work.

**Abstract:** The negative impacts of extreme heat and drought on olive plants have driven the quest for mitigation approaches based on the use of biostimulants, which have proved to be effective in contrasting environmental stresses. The aim of our study was to evaluate the effectiveness of six biostimulants in mitigating high temperature and water stress in young olive trees in terms of vegetative and eco-physiological parameters as well as bioactive compound content. Biostimulants based on glycine betaine and macro- and micro-algae effectively protected the plants from abiotic stress by improving their eco-physiological and vegetative parameters. At the end of the growing season, olive plants were experiencing water deficit which had built up through the summer months. At this time, the glycine betaine-treated plants had a three-fold higher stomatal conductance compared with the control, while plants sprayed with the seaweed mix had a relative water content 33% higher than the control. The kaolin treatment resulted in higher total phenolics and antioxidant activities (DPPH, FRAP and ABTS) in water stress conditions and caused an increase of 238.53 and 443.49% in leaves total polyphenols content in 100% and 50% water regime, respectively. This study showed the effectiveness of biostimulants in mitigating the damage from abiotic stress on young olive trees, by improving some vegetative, eco-physiological and leaf nutraceutical parameters. Further studies are needed to test the efficiency of these biostimulants in open field conditions on olive trees in full production.

**Keywords:** *Olea europaea* L.; *Trichoderma*; *Ascophyllum nodosum*; *Laminaria digitata*; pinolene; phenolic profile; high resolution mass spectrometry; antioxidant activity

## 1. Introduction

Olive (*Olea europaea* L.) is one of the most important fruit trees of the Mediterranean region with an enormous economic and ecological value. In its climatic region, olive is often subject to drought periods during the warm season; nevertheless, it is characterized by high morphological and physiological adaptation capacities [1]. Water stress was reported to induce in olive plants the activation of antioxidant enzyme systems such as ascorbate peroxidase, catalase and superoxide dismutase [2]. Moreover, the activation of the phenyl-propanoid biosynthetic pathway leading to the accumulation of phenolic compounds is a well-known metabolic response to water deficit as well as to other environmental stresses [2]. Such metabolic responses of the plants to unfavorable environmental conditions play a key role in preventing cellular damage caused by oxidative stress. However, the constitutive tolerance to water deficit alone is not sufficient to protect olive trees from the combined effects of extreme heat waves, water stress and high irradiance, which are all linked to

climate change. For this reason, the application of biostimulants as a sustainable practice to mitigate the negative impact of environmental stresses and to meliorate or maintain plant productivity has become increasingly popular over the past decade [3]. In this respect, different categories of biostimulants have been proved effective to minimize the effects of abiotic stresses on plants. Among these, plant-derived biostimulants are particularly interesting due to their biocompatibility and low environmental impact [4]. The osmolyte glycine betaine is widely studied and its effectiveness towards numerous sources of stress such as drought, cold or salinity, were reported [5]. Kaolin clay reflective particles are also well known as an effective tool to reduce the effects of abiotic stress on crop performance by affecting the plant at the morphological, physiological and biochemical levels [6]. In this case, however, contrasting results were reported, depending on the species or even the genotype [6,7]. Pinolene, is a therpenic polymers (Di-1-p-menthene), which is inactive from a biochemical point of view and it is mainly used in crops to limit leaf water loss since it acts as film-forming antitranspirant [8,9]. Pinolene has been proposed as an alternative to defoliation in hot climates [10–13]. Among the algal extracts, foliar applications of a biostimulant based on *Ascophyllum nodosum* and *Laminaria digitata* were shown to be involved in the regulation of secondary metabolism, resulting in improved fruit quality and nutritional value of apples cv. "Annurca" [14]. In addition, Kusvuran [15] reported that under drought stress, the foliar application of micro-algae such as *Chlorella vulgaris*, significantly improved secondary metabolites, such as polyphenolic compounds and antioxidant enzyme activities. Furthermore, Graziani et al. [14] reported that treatments with micro-algae on apple trees cv. "Annurca" also improved post-harvest fruit conservation by preserving the nutritional quality, in terms of polyphenols content after 120 days of cold storage. Fungi of the *Trichoderma* genus are widely studied and commonly used as biostimulants in agriculture. Rudresh et al. [16] showed that a mixture of *T. harzianum, T. viride* and *T. virens* increased plant biomass and nutrient uptake. Some reports showed beneficial effects of *Thricoderma* on the alleviation of salt stress effects [17,18]. Dini and co-workers recently reported that *Trichoderma* strains increase phenolics concentration both in olive leaves and in oil, as a result of improved plant nutrient uptake and enhanced nitrogen use efficiency [19].

On the basis of this knowledge, we tested the application of six different biostimulant treatments as an approach to overcome/balance the negative impact of high temperatures and water deficit on young olive trees. In this study we monitored agronomic and eco-physiological parameters as well as the antioxidant activities and the phenolic composition of leaf extracts. Such results would be of interest to olive producers, presenting solutions for stress mitigation and crop conservation.

## 2. Materials and Methods

### 2.1. Plant Material, Biostimulants Treatments and Experimental Design

The trial was conducted in a greenhouse at the Department of Agriculture of the University of Naples Federico II, Portici—Italy, between the end of May and the end of September 2020. Two-year-old potted olive trees of the cultivar "Salella", were grown in 5 L pots containing a substrate made up of sand:peat:clayey soil (1:1:1, *v/v/v*). At the beginning of the experiment, all olive plants had homogeneous vegetative characteristics and 50 g/pot of "Nitrophoska Gold" (a slow-release fertilizer based on N (15%), $P_2O_5$ (9%), $K_2O$ (15%) supplemented with micronutrients) by COMPO EXPERT Italia Srl (Cesano Maderno, Monza and Brianza, Italy) was mixed with the substrate.

The experimental design was based on seven biostimulant treatments:

(1) Control (C) plants only treated with water, no biostimulant applied.
(2) *Trichoderma* based product (TR), "Trianum-P" by Koppert Biological Systems (Koppert Italy, Bussolengo, VR—Italy), with active ingredient *Trichoderma harzianum* Rifai strain T-22 (also known as KRL-AG2*). The product was applied to the root system by irrigation at the dose of 6.67 g/L of water.

(3)  Micro-Algae-based product (MA), "AgriAlgae® Biologico Originale" by AgriAlgae® (Madrid, Spain), a biological biostimulant. The product was applied to the root system by irrigation at the dose of 6.67 g/L of water.

(4)  Seaweed based product (P), "Seaweed Mix®" by L. Gobbi Srl (Campo Ligure, Genoa, Italy), made of *Ascophyllum nodosum* and *Laminaria digitate* extract. The product was applied to the root system by irrigation at the dose of 4 mL/L of water.

(5)  Glycine betaine based product (B), "BIO-HELP" by Biolchim SPA (Bologna, Italy), a bio-promoter of resistance to environmental stress. The product was applied to the root system by irrigation at the dose of 10 g/L of water.

(6)  Kaolin (K), "Manisol" by Manica S.p.A (Rovereto, Italy). The product was applied as foliar spray at the dose of 40 g/L of water.

(7)  A water emulsifiable organic concentrate of di-1-p-menthene ($C_{20}H_{34}$) (V), "Vapor Gard®" by BIOGARD® (Bergamo, Italy), a terpenic polymer also known as pinolene. The product was applied as foliar sppray at the dose of 10 mL/L of water.

All biostimulants were applied five times during the growing season at 20 day intervals. Olive trees were divided into two groups, corresponding to two watering regimes: 100% and 50% of the evapotranspiration (ET). Evapotranspiration was calculated on the well-watered plants of the 100% watering regime group using a gravimetric method as follows: every two days the pots were weighed up and water loss by evapotranspiration was calculated.

Subsequently, 100% or 50% of water loss, corresponding to X mL or X/2 mL water per pot, respectively, was restored by drip irrigation. The drip irrigation system was controlled by a programmable timer and it was powered by an electric pump, feeding water to drippers at a flow rate of 2 L/h. One or two drippers were installed into each pot for irrigation at 50% or 100%, respectively.

A total of 14 treatments were compared based on a factorial combination of seven biostimulant treatments (including control) and two irrigation regimes (100% and 50%). The treatments were arranged in a randomized split-plot design with irrigation levels as main factor and biostimulants as sub-factor. Each treatment consisted of 10 plants.

### 2.2. Determination of Vegetative and Eco-Physiological Parameters of Leaves

On fully developed leaves, the stomatal conductance was measured using a Porometer (Li-1600 Steady State Porometer, TR. Turoni Srl, Forli, Italy) at 12:00 a.m. The leaf SPAD index was measure with a chlorophyll meter SPAD-502 (Konica-Minolta, Osaka, Japan). The leaf relative water content (RWC) was calculated following the previously described procedure [7] according to the formula:

$$RWC\ (\%) = ((fw - dw)/(rw - dw)) * 100$$

The number of leaves per plant was recorded at the beginning and at the end of the experiment in order to calculate the increase in leaf number. Leaf area per plant was measured using imageJ software version 1.50 (Wayne Rasband, National Institute of Health, Bethesda, MD, USA) at the end of the experiment. Leaf dry weight was determined by drying sub-samples in a forced air oven until constant weight was reached. Polyphenolic content and antioxidant activity assays were determined on lyophilized leaves.

The eco-physiological measurements were carried out in June (one month after the first biostimulant application) and in September (at the end of the growing season), taking six measurements per treatment.

### 2.3. Chemicals Analyses and Ultrasound-Assisted Extraction of Polyphenolic Compounds

All standards for the analysis were supplied by Sigma Aldrich St. Louis, MO, USA, while hydroxytyrosol was purchased from Indofine (Hillsborough, NJ, USA), secologanoside from ChemFaces Biochemical Co., Ltd. (Wuhan, China) and oleuropein form Extrasynthese (Genay, France). Acetonitrile and water (LC-MS grade) were acquired from

Carlo Erba reagents (Milan, Italy), whereas acetic acid (98–100%) was purchased from Fluka (Milan, Italy).

Lyophilized samples were extracted using the method reported in the literature [20] with few modifications. In particular, 0.3 gr of dried sample were extracted with 15 mL of methanol/water (80:20 *v/v*, 0.1% formic acid) by sonication at room temperature for 15 min. Samples were centrifuged to 4000 rpm at 4 °C for 10 min, and the pellet was extracted in the same way. The supernatants were collected, filtered through 0.45 mm nylon syringe membranes and then used for high-resolution mass spectrometry analysis and antioxidant activity assays.

*2.4. UHPLC-HRMS Analysis of Polyphenolic Compounds*

An Ultra-High-Pressure Liquid Chromatograph (UHPLC, Dionex UltiMate 3000, Thermo Fisher Scientific, Waltham, MA, USA) coupled with a Q-Exactive Orbitrap mass spectrometer (UHPLC, Thermo Fischer Scientific, Waltham, MA, USA) was used to investigate the quali-quantitative profile of polyphenolic compounds applying conditions reported in our previous work [19]. An Accucore aQ 2.6 μm (100 × 2.1 mm) column (Thermo Scientific, Waltham, MA, USA) was applied for chromatographic separation of polyphenols with a column temperature set at 30 °C. The mobile phase consisted of water containing 0.1% of acetic acid (eluent A) and acetonitrile (eluent B). Polyphenolic compounds were eluted using the following gradient program: 0–5 min 5% B, 5–25 min 5–40% B, 25–25.1 min 40–100% B, 25.1–27 min 100% B, 27–27.1 min 100–5% B, 27.1–35 min 5% B. The flow rate was 0.4 mL min$^{-1}$ and the injection volume was 5 μL. The mass spectrometer was operated in negative ion mode (ESI–) setting two scan events (Full ion MS and All ion fragmentation, AIF) for all compounds of interest. Full scan data were acquired setting a resolving power of 35,000 FWHM (full width at half maximum) at m/z 200. The key parameters were as follows: spray voltage −2.8 kV, sheath gas flow rate 35 arbitrary units, auxiliary-gas flow rate, 10 arbitrary units, capillary temperature 275 °C, auxiliary gas heater temperature 350 °C, S-lens RF level 50. For the scan event of AIF, the resolving power was set at 17,500 FWHM, the collision energies were 10, 20, and 45 eV, and the scan range was m/z 80–1200. Data acquisition and processing were performed with Quan/Qual Browser Xcalibur software, v. 3.1.66.10 (Xcalibur, Thermo Fisher Scientific, Waltham, MA, USA).

*2.5. Antioxidant Activity: ABTS Assay*

Determination of the ABTS free radical scavenging activity was carried out following the method described by Re et al. [21]. Briefly, 44 μL of aqueous potassium persulfate (2.45 mM) were added to 2.5 mL of aqueous ABTS (7 mM) and incubated in the dark at room temperature for 12–16 h. The ABTS solution was diluted with ethanol (1:88) to obtain an ABTS radical working solution with an absorbance value of 0.75 ± 0.050 at 734 nm. The assay was performed by adding 100 μL of properly diluted sample to 1 mL of ABTS radical working solution and the absorbance was monitored after 2.5 min at 734 nm. Results were expressed as Trolox equivalent antioxidant capacity (TEAC, mmol Trolox equivalents kg$^{-1}$ dry weight of leaves). All determinations were performed in triplicate.

*2.6. Antioxidant Activity: DPPH Assay*

The DPPH assay was carried out according to the procedure reported by Brand-Williams et al. [22] with minor modifications. Briefly, methanolic DPPH radical working solution was prepared diluting methanolic DPPH (4 mg in 10 mL) with methanol, until an absorbance value of 0.900 ± 0.020 at 517 nm. For the assay, 200 μL of sample were added to 1 mL of radical working solution and the absorbance value was monitored after 10 min. The results were expressed as Trolox equivalents antioxidant capacity (TEAC, mmol Trolox equivalent kg$^{-1}$ of dry weight of leaves). All determinations were performed in triplicate.

### 2.7. Antioxidant Activity: FRAP Assay

The FRAP assay was conducted according to the method reported by Benzie and Strain [23] with slight adjustments as mentioned in Formisano et al. [24].

Briefly, the FRAP reagent was made up of 10 μM TPTZ in 40 μM HCl, 20 μM of aqueous $FeCl_3$ and acetate buffer (300 μM, pH 3.6) at 1:1:10 (*v/v/v*). Sample solutions, properly diluted, (10 μL) and FRAP reagent (300 μL) were mixed and the absorbance was monitored at 593 nm after 10 min. The results were expressed in mmol Trolox® $Kg^{-1}$ dry weight (dw). The results were corrected for dilution and expressed as Trolox® equivalent antioxidant capacity (TEAC, mmol Trolox equivalents $Kg^{-1}$ dry weight of leaves). All determinations were performed in triplicate.

### 2.8. Total Polyphenol Content: FOLIN Test

Total phenolics were determined according to a Folin-Ciocalteu procedure with slight changes [25]. Briefly, 125 μL of diluted extract or blank was mixed with 500 μL of deionized water and 125 μL of the Folin-Ciocalteu reagent for 6 min at room temperature. Subsequently, 1.25 mL of 7.5% of sodium carbonate solution and 1 mL of deionized water were added in the mixture. The absorbance at 760 nm after 90 min. of incubation in the dark was measured. Concentrations of total phenolic were expressed in terms of mg of gallic acid equivalents (GAE) per gram dry weight (DW), based on a calibration curve (R2 > 0.993) that was computed over a dynamic range 0.05–2.5 g/L gallic acid. Each extract was analyzed in triplicate.

### 2.9. Statistical Analysis

All data were subjected to analysis of variance (ANOVA). Duncan's multiple range test (DMRT) was performed for means separation of each of the measured variables at $p = 0.05$. A principal component analysis (PCA) was executed on vegetative, physiological and bioactive parameters at the end of growing season to detect the interrelationship. The PCA results are shown as a biplot to highlight the interaction between samples and variables. Samples are displayed as points while variables are displayed as vectors. A correlation analysis between the total phenolic content (Folin) and each of the antioxidant capacity assays (ABTS, DPPH and FRAP) was performed. The statistical package XLStat Version 2013 (New York, NY, USA) was implemented for all the analyses.

## 3. Results and Discussion

### 3.1. Vegetative and Eco-Physiological Parameters

Figure 1 shows the temperature values recorded in the greenhouse during the growing season, from May to September. The young olive trees were exposed to prolonged thermal stress: the maximum daily temperature ranged between 23.42 °C (4 June) and 51.74 °C (2 June), while the minimum daily temperature ranged between 10.20 °C (31-May) and 25.01 °C (2 June). The average daily temperatures were between 19.44 °C (5 June) and 36.97 (2 June).

In Table 1, the vegetative parameters determined on the leaves are reported: leaf dry weight (g), leaf number and leaf area ($cm^2$). A significant interaction emerged between the two tested factors (watering regime × biostimulant treatment). Leaf dry weight and leaf area were higher in plants at the 100% water regime and were not influenced by the different biostimulant treatments. Leaf dry matter did not show significant differences between the various treatments compared with the control (C) in both watering regimes. The two watering regimes also affected the production of new leaves: plants in the group at the 100% watering regime produced 89% new leaves, compared with 48% new leaves at the 50% watering regime. Therefore, the plant group exposed to water deficit produced 45% less new leaves compared to the well-watered group. These results are consistent with previous literature on the effects of water deficit: vegetative growth is closely related to plant water status since a loss of turgor impairs cell expansion and results in reduced plant growth in terms of plant height, leaf area, dry weight and other vegetative parameters [26,27]. Compared with the untreated control under the 50% watering regime, plants treated with

glycine betaine (B) had a significantly higher leaf number (+44%) and 56% larger leaf area, while the micro-algae (M) and seaweed (P) treated plants had a 26% and 44% larger leaf area, respectively. Our results are in agreement with previous studies reporting that glycine betaine improved plant resistance to high temperature stress [28]. Moreover, Alia et al. [29] reported that tolerance to abiotic stress during the imbibition and germination of seeds, as well as during the growth of young seedlings was enhanced in transformed *Arabidopsis thaliana* accumulating glycine betaine. The exogenous application of glycine betaine was also reported to positively affect plant growth and final crop yield under drought stress [30,31]. Seaweed extracts are employed in agriculture for their beneficial effects on plant growth, root development, mineral nutrition and fruit setting as well as improved resistance to abiotic stresses (drought, salinity and temperature), pests and diseases as recently reviewed by Mukherjee and Patel [32].

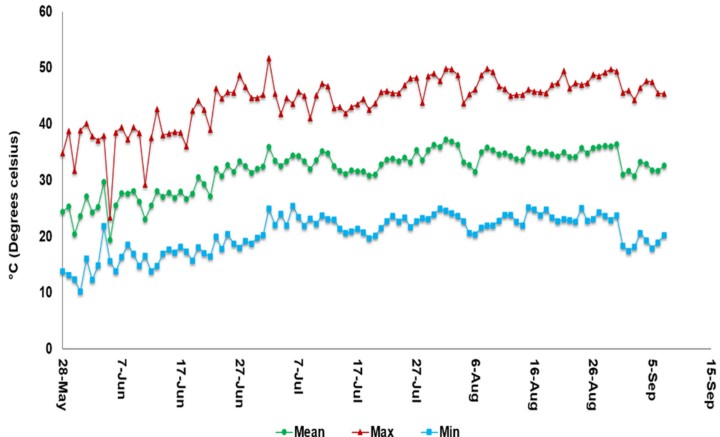

**Figure 1.** Temperature trend recorded during the growing season from May to September under greenhouse conditions: maximum, minimum, and mean temperature (°C).

**Table 1.** Leaf dry matter (g), increase in leaf number and leaf area ($cm^2$) of olive trees grown in a greenhouse under optimal irrigation (100% irrigation) and limited water availability (50% irrigation) treated with six different biostimulants: C = control, TR = *Trichoderma*, M = Micro-Algae, P = Seaweed mix, K = Kaolin, B = Glycine betaine, V = Pinolene.

| Treatments | Leaf Dry Matter (g) | Increase Leaves Number | Leaf Area ($cm^2$) |
|---|---|---|---|
| **100%** | | | |
| C | 16.97 ± 0.61 a | 83.70 ± 7.01 abc | 770.29 ± 25.56 a |
| TR | 18.37 ± 0.78 a | 88.70 ± 5.29 abc | 729.59 ± 19.18 a |
| M | 18.06 ± 0.58 a | 86.70 ± 8.03 abc | 758.90 ± 27.87 a |
| P | 17.20 ± 0.97 a | 82.80 ± 6.93 abc | 704.16 ± 25.26 a |
| B | 16.38 ± 0.53 a | 102.00 ± 8.17 a | 736.15 ± 25.96 a |
| K | 17.93 ± 0.71 a | 77.50 ± 7.26 bc | 759.66 ± 41.60 a |
| V | 18.57 ± 0.81 a | 94.50 ± 4.81 ab | 723.25 ± 28.17 a |
| **50%** | | | |
| C | 10.20 ± 0.53 bcd | 49.40 ± 5.47 ef | 361.81 ± 16.27 ef |
| TR | 8.74 ± 0.63 d | 39.70 ± 5.48 ef | 330.14 ± 11.49 f |
| M | 9.87 ± 0.69 cd | 49.20 ± 10.68 ef | 454.09 ± 30.06 cd |
| P | 11.61 ± 0.85 bc | 45.20 ± 4.88 ef | 522.28 ± 26.05 bc |
| B | 10.77 ± 0.53 bcd | 70.90 ± 3.76 cd | 562.84 ± 34.43 b |
| K | 12.08 ± 0.52 b | 30.40 ± 5.46 f | 437.72 ± 19.46 cde |
| V | 12.32 ± 0.69 b | 54.20 ± 10.76 de | 515.79 ± 34.02 cde |
| **Significance** | | | |
| W | *** | *** | *** |
| T | ns | * | ns |
| W × T | *** | *** | *** |

Values are mean ± standard error. Asterisks indicate significant effect of limited water availability (W), biostimulants treatments (T) and their interaction (W × T) according to ANOVA (ns = not significant; * = $p < 0.05$; *** = $p < 0.001$). Different letters indicate significant differences based on Duncan's test ($p = 0.05$).

The eco-physiological parameters (stomatal conductance, SPAD and RWC) measured at the beginning (1 month following the first biostimulant application) and at the end of the growing season are reported in Table 2. For each of the three parameters, significant interactions emerged among the experimental factors W × T × S (water regime × treatments × measurement time).

**Table 2.** Stomatal conductance (mol m$^{-2}$ s$^{-1}$), SPAD and RWC (%) values in leaves of olive trees grown in a greenhouse under optimal irrigation (100% irrigation) and limited water availability (50% irrigation) treated with several biostimulants: C = control, TR = *Trichoderma*, M = Micro-Algae, P = Seaweed mix, K = Kaolin, B = Glycine betaine, V = Pinolene; at the start (Time 1: T1; 1 month after the first biostimulant application) and at the end (Time 2: T2) of the experiment.

| | Time 1 | | Time 2 | |
|---|---|---|---|---|
| **Treatments** | **100%** | **50%** | **100%** | **50%** |
| **Stomatal conductance (mol m$^{-2}$ s$^{-1}$)** | | | | |
| C | 0.534 ± 0.01 bc | 0.165 ± 0.00 efgh | 0.148 ± 0.02 efgh | 0.064 ± 0.00 fgh |
| TR | 0.538 ± 0.04 bc | 0.120 ± 0.02 efgh | 0.253 ± 0.04 def | 0.036 ± 0.01 gh |
| M | 0.233 ± 0.00 def | 0.686 ± 0.10 b | 0.616 ± 0.12 b | 0.257 ± 0.02 de |
| P | 0.276 ± 0.02 def | 0.118 ± 0.01 efgh | 0.389 ± 0.19 cd | 0.030 ± 0.01 h |
| B | 0.875 ± 0.01 a | 0.539 ± 0.04 bc | 0.557 ± 0.06 b | 0.271 ± 0.03 de |
| K | 0.193 ± 0.03 efgh | 0.231 ± 0.01 def | 0.054 ± 0.00 fgh | 0.057 ± 0.00 fgh |
| V | 0.195 ± 0.05 efgh | 0.211 ± 0.03 efg | 0.149 ± 0.00 efgh | 0.221 ± 0.03 def |
| **SPAD** | | | | |
| C | 75.57 ± 1.20 ghij | 76.29 ± 0.77 fghi | 73.06 ± 1.27 j | 76.50 ± 0.74 fghi |
| TR | 74.39 ± 1.35 hij | 75.23 ± 0.82 ghij | 74.20 ± 1.52 hij | 77.05 ± 1.13 efghi |
| M | 80.88 ± 0.83 bc | 76.82 ± 0.99 efghi | 78.93 ± 0.72 cdef | 80.00 ± 0.95 bcde |
| P | 78.23 ± 1.00 cdefg | 77.83 ± 0.81 cdefg | 75.68 ± 0.85 fghij | 79.79 ± 1.02 bcde |
| B | 80.52 ± 1.08 bcd | 82.63 ± 0.73 ab | 80.54 ± 0.85 bcd | 78.87 ± 0.75 cdef |
| K | 76.08 ± 1.05 fghij | 74.41 ± 1.17 hij | 78.03 ± 0.65 cdefg | 84.42 ± 0.64 a |
| V | 78.00 ± 0.85 cdefg | 74.04 ± 0.89 ij | 77.45 ± 1.20 defgh | 76.93 ± 0.73 efghi |
| **RWC%** | | | | |
| C | 79.87 ± 2.60 ab | 75.14 ± 4.20 abcd | 60.87 ± 3.46 fgh | 55.24 ± 1.71 ghi |
| TR | 79.90 ± 1.92 ab | 76.33 ± 3.22 abc | 65.56 ± 1.64 def | 68.56 ± 6.69 cdef |
| M | 81.09 ± 1.45 a | 73.82 ± 2.49 abcd | 63.11 ± 4.32 efg | 50.95 ± 1.06 i |
| P | 75.67 ± 3.81 abcd | 78.15 ± 3.18 abc | 62.45 ± 1.82 fgh | 73.32 ± 4.32 abcd |
| B | 74.51 ± 2.83 abcd | 80.56 ± 3.20 ab | 72.96 ± 1.71 abcde | 61.22 ± 2.18 fgh |
| K | 74.87 ± 0.72 abcd | 77.82 ± 1.11 abc | 67.82 ± 2.48 cdef | 55.54 ± 0.92 ghi |
| V | 74.25 ± 4.94 abcd | 83.66 ± 1.69 a | 70.06 ± 4.08 bcdef | 52.55 ± 4.62 hi |
| | Stomatal conductance (mol m$^{-2}$ s$^{-1}$) | | SPAD | RWC (%) |
| W | ns | | ns | ns |
| T | *** | | *** | ns |
| S | ns | | ns | *** |
| W × T × S | *** | | *** | *** |

Values are means ± standard error. Asterisks indicate significant effect of limited water availability (W), biostimulant treatment (T), time of measurements (S) and their interaction (W × T × S) according to ANOVA (ns = not significant; *** = $p < 0.001$). Different letters indicate significant differences based on Duncan's test ($p = 0.05$).

The highest stomatal conductance value (Table 2) was recorded at Time 1 in plants at the 100% watering regime treated with glucine betaine (0.875 mol m$^{-2}$ s$^{-1}$). At the same time, plants treated with glycine betaine (B) and with micro-algae (M) had the highest stomatal conductance values within the group grown at the 50% watering regime. Stomatal conductance of B and M treated plants at the 50% watering regime was comparable to control plants at the 100% watering regime. The effectiveness of the B an M treatments in maintaining an high stomatal conductance was confirmed at the second measurement (Time 2), when the measured values were 0.557 and 0.616 mol m$^{-2}$ s$^{-1}$ for the B and the M treatments, respectively, at the 100% watering regime. Stomatal conductance of B and M

treated plants was significantly higher than the control (0.148 mol m$^{-2}$ s$^{-1}$). A similar effect was recorded in plants grown at the 50% watering regime, as plants treated with B and M biostimulants maintained a higher stomatal conductance compared with the control.

Plants treated with the B and the M biostimulants also had significantly higher SPAD index values compared with the control, both at Time 1 and at Time 2 in the case of 100% watering regime. The SPAD index of M treated plants was 7% and 8% higher than Control at Time 1 and at Time 2, respectively, while B treated plants had SPAD index 7% and 10% higher than Control at Time 1 and Time 2, respectively. Among plants at the 100% watering regime, K and V treated plants at Time 2 also had 7% and 6% higher SPAD index than the control, respectively. Among plants grown in water deficit conditions (50% watering regime), the SPAD index for B treated plants was 8% higher than the control at Time 1, while M, P and K treated plants had aSPAD index 5%, 4% and 10% higher than the control, respectively, at Time 2. Our results are in agreement with previous studies reporting that the application of seaweed extracts increased chlorophyll content in leaves [33–36]. Moreover, treatments with seaweed extracts were also reported increase the leaf area as demonstrated in our study. Interestingly, the M and B treatments significantly increased the SPAD index: this may result from a protective effect of seaweed extracts and betaines on chlorophyll as in the reported case of glycine betaine delaying the loss of photosynthetic activity of isolated chloroplasts [37].

The leaf RWC values (Table 2) reflect the water status of the plant: accordingly, higher RWC values were measured at the beginning of the experiments (Time 1) both at 100% and at 50% watering regimes, while lower values were measured after prolonged stress at the end of summer (Time 2). At Time 1, no significant differences emerged between the control and the different biostimulant treatments at either 100% or 50%. Contrastingly, at Time 2 among the 100% watering regime group only plants treated with glycine betaine (B) had a significantly higher (+20%) RWC compared to the control, while in the case of plants at the 50% watering regime, TR and P treated plants had 24% and 33% higher RWC than the control, respectively. The reported effect of the TR treatment in maintaining the plant water status is in agreement with Shukla et al. [38], who showed a significant decrease in the RWC in response to drought stress in untreated *Triticum aestivum* plants, while colonized plants with drought-tolerant *Trichoderma* isolates were able to retain water. However, osmotic adjustment was higher in *Trichoderma*-colonized wheat plants compared to an untreated control and the degree of osmotic adjustment increased with the intensity of drought. Further, another study found that in *P. eugenioides*, the application of seaweed extracts under 100% ET irrigation conditions had no significant effects on improving RWC mean values; however, under water stress conditions (50% ET) the RWC remained significantly higher than the untreated control [39]. In agreement with our results, when exposed to water deficit plants lose water over time with a gradual reduction in transpiration rate [40]. Our results are in line with a previous study [41] showing that a treatment with glycine betaine increased RWC in stressed plants. These data suggested that glycine betaine could increase the plant hydraulic conductivity, enhancing the water flow from roots to shoots and eventually increasing RWC and transpiration rate under stress conditions [41].

### 3.2. Polyphenolic Compounds Analysis by UHPLC-Q-Orbitrap HRMS

The phenolic composition (12 phenolic compounds and their formula) gathered from the UHPLC-HRMS analysis are presented in Table S1, whereas a typical full-scan MS chromatogram of olive leaves extract is reported in Supplementary Figure S1.

Table 3 shows the quali-quantitative polyphenolic profile of olive leaves in control and biostimulant treated samples at two different water regimes (100% and 50% of evapotraspiration).

**Table 3.** Phenolic profiles and total phenolic composition in leaves of olive tree grown in a greenhouse under optimal irrigation (100% irrigation) and limited water availability (50% irrigation) treated with several biostimulants: C = control, TR = *Trichoderma*, M = Micro-Algae, P = Seaweed mix, K = Kaolin, B = Glycine betaine, V = Pinolene. Concentrations were expressed as mg/g dw.

| Polyphenols | C | | TR | | M | | P | |
|---|---|---|---|---|---|---|---|---|
| | 100% | 50% | 100% | 50% | 100% | 50% | 100% | 50% |
| hydroxytirosol glucoside | 2183.11 e | 1017.86 g | 4978.55 d | 1905.95 ef | 1959.59 ef | 1601.89 efg | 5630.96 c | 1664.50 efg |
| vanillic acid | 5.78 bcde | 3.85 g | 7.01 b | 5.41 cdef | 6.03 bcd | 4.89 defg | 6.53 bc | 4.26 fg |
| coumaric acid | 4.82 b | 2.24 ef | 3.14 cd | 3.39 c | 2.49 def | 2.56 def | 4.86 b | 2.04 f |
| ferulic acid | 3.91 c | 1.43 fgh | 4.84 b | 1.82 efg | 2.28 e | 1.42 fgh | 4.15 c | 1.31 gh |
| luteolin rutinoside | 7.37 de | 8.62 bc | 8.05 cd | 9.01 bc | 7.33 de | 9.16 b | 7.23 de | 7.57 de |
| verbascoside | 156.91 e | 118.14 f | 504.48 b | 715.50 a | 403.05 d | 472.92 c | 28.96 g | 25.11 g |
| oleuropein | 629.34 gh | 750.22 fg | 966.74 d | 1274.20 c | 1058.80 d | 813.96 ef | 912.37 de | 579.43 h |
| ligstroside | 54.01 g | 58.26 g | 77.32 f | 108.24 cd | 67.40 fg | 91.11 e | 113.72 c | 102.90 cde |
| pinoresinol | 0.39 hi | 0.34 i | 0.54 ef | 0.52 ef | 0.71 d | 0.80 c | 0.44 gh | 0.33 i |
| luteolin | 205.00 cde | 157.81 h | 163.93 gh | 175.82 fg | 199.02 cde | 195.64 de | 201.15 cde | 211.23 cd |
| oleuropein aglycone | 54.67 h | 38.27 i | 67.72 g | 95.61 e | 82.34 f | 83.22 f | 23.06 l | 27.80 l |
| secologanoside | 15.08 f | 23.88 de | 45.76 b | 58.54 a | 20.16 def | 21.27 def | 17.37 ef | 15.61 f |
| Total polyphenols | 3322.40 de | 2180.93 f | 6828.03 b | 4354.01 c | 3809.17 cd | 3298.83 de | 6950.79 b | 2642.09 ef |

| Polyphenols | K | | B | | V | | Significance | | |
|---|---|---|---|---|---|---|---|---|---|
| | 100% | 50% | 100% | 50% | 100% | 50% | T | W | W × T |
| hydroxytirosol glucoside | 8215.27 b | 8915.23 a | 1540.62 efg | 1562.68 efg | 980.91 g | 1251.98 fg | *** | ns | *** |
| vanillic acid | 9.06 a | 5.30 def | 8.68 a | 6.42 bc | 4.59 efg | 3.67 g | *** | *** | *** |
| coumaric acid | 5.70 a | 3.51 c | 2.08 f | 2.05 f | 3.51 c | 2.91 cde | ** | *** | *** |
| ferulic acid | 7.19 a | 1.79 efgh | 1.29 gh | 1.19 h | 1.98 ef | 3.06 d | * | *** | *** |
| luteolin rutinoside | 10.19 a | 10.40 a | 6.92 e | 8.66 bc | 7.25 de | 8.97 bc | *** | *** | *** |
| verbascoside | 381.39 d | 472.28 c | 23.28 g | 31.35 g | 31.51 g | 35.97 g | *** | ns | *** |
| oleuropein | 1868.74 a | 1657.18 b | 912.29 de | 1006.04 d | 575.93 h | 955.20 de | *** | ns | *** |
| ligstroside | 266.16 a | 241.43 b | 103.29 cde | 94.61 de | 67.79 fg | 96.12 de | *** | ns | *** |
| pinoresinol | 0.57 e | 0.64 d | 1.19 b | 1.43 a | 0.46 fg | 0.54 e | *** | ns | *** |
| luteolin | 217.60 c | 258.80 b | 315.03 a | 324.19 a | 191.02 ef | 176.19 fg | *** | ns | *** |
| oleuropein aglycone | 230.89 b | 242.61 a | 106.02 d | 122.09 c | 45.91 hi | 49.40 h | *** | ns | *** |
| secologanoside | 34.06 c | 39.67 bc | 13.83 f | 14.12 f | 13.36 f | 25.23 d | *** | ns | *** |
| Total polyphenols | 11246.82 a | 11848.83 a | 3034.54 de | 3174.82 de | 1924.22 f | 2609.15 ef | *** | ns | *** |

Values are mean and asterisks indicate significant effect of limited water availability (W), biostimulants treatments (T) and their interaction (W × T) according to ANOVA (ns = not significant; * = $p < 0.05$; ** = $p < 0.01$; *** = $p < 0.001$). Different letters indicate significant differences based on Duncan's test ($p = 0.05$).

### 3.3. Antioxidant Activity of Polyphenolics Extracts

Oleuropein and hydroxytirosol glucoside were the two main phenolic compounds detected. By contrast, the presence of tirosol was not observed. Hydroxytyrosol glucoside represented the main component in the olive leaves of well-watered and stressed plants, reaching an average 58.6% on the sum of all polyphenolic compounds. It was also noteworthy that the concentrations of this compound referred to in this work were in general higher than those illustrated by other researchers [20]; nonetheless, these latters used different cultivars, the sampling conditions reported in those works were different, and these factors could considerably alter levels of this metabolite. According to the literature data, oleuropein was also well represented in all leaf samples, representing on average 24.5% of the total compound concentration.

Verbascoside, ligstroside and oleuropein aglycone were also present at significant concentrations, ranging between 16.43–0.42%, 3.90–1.13% and 3.49–9.33% of total polyphenolic compounds, respectively. Among flavonoids, luteolin was the most abundant compound exhibiting levels between 10.38 and 1.93% of total polyphenols. Coumaric, ferulic and vanillic acid were found in minor concentrations as well as luteolin rutinoside, pinoresinol and secologanoside. In accordance with literature data [42] the concentration of hydroxytirosol glucoside showed higher values in leaves from fully irrigated plants compared to water stressed leaves for untreated leaves and for all other treatments with the exception of Kaolin and Pinolene, which generated in the leaves a higher concentration of this compound under water stress conditions.

The concentration of oleuropein was shown to be dependent on the type of biostimulant treatment. In fact, a higher level this metabolite was observed in the leaves of plants

subjected to water stress in the case of the untreated samples and after foliar the application of *Trichoderma*, glycine betaine and pinolene. Higher levels of oleuropein were found in the leaves of fully irrigated plants following treatment with micro-algae, seaweed mix and kaolin. An increased concentration of oleuropein in olive leaves subjected to drought stress was reported by Petridis et al. [2] and by Talhaoui et al. [20], especially in drought stress of control trees.

The water stress condition resulted in an increase in verbascoside when *Trichoderma*, seaweed mix, and kaolin were applied to the plants, and a decrease in hydroxycinnamic acids in all the examined treatments. Water deficit, however, did not affect minor polyphenolic compounds. Overall, leaf polyphenol content decreased as a consequence of water deficit, while no decrease was observed in plants treated with kaolin, glycine betaine and pinolene (Table 3). In accordance with literature data [42] the water deficit induced by the 50% the watering regime caused a reduction in polyphenol content, especially in the case of treatment with seaweed extracts (P treatment) which led to a reduction in polyphenol content of about 61.98%. A slight decrease of polyphenolic content was observed with micro-algae treatment (about 13.49%), while in the control and *Trichoderma* treatments, a similar decrease of about 34.35% and 36.23%, respectively, was found.

Plant stress response is related to bioactive metabolite arrangements which are dependent both on the plant (species and cultivar) and on the nature and intensity of the stress factor. Polyphenolic compounds such as flavonoids, secoiridoids and hydroxycinnamic acid derivatives are involved in the plant stress defense as they act as antioxidants useful to counteract oxidative stress [42]. Therefore, the decrease of such compounds under stress conditions may be related to the defense-related functions of phenolic compounds. Interestingly, in the case of treatments with kaolin, glycine betaine and pinolene, no significant differences between irrigation regimes were observed. Therefore, these last treatments support the implementation of agronomic practices to mitigate the negative consequences of water stress. In addition, the positive effects of the kaolin treatment on polyphenols biosynthesis could be related to the up regulation of gene transcription encoding chalcone synthase and phenylalanine ammonia lyase (PAL), as previously described in grape berries subjected to drought and heat stress [43]. On the other hand, Denaxa et al. [44] reported that water-stressed olive leaves treated with kaolin exhibited similar lipid oxidation, evaluated by measuring thiobarbituric acid reactive substances (TBARS), to those under fully irrigated conditions. This suggests that antioxidant defense systems under drought were sufficient and effective to counteract ROS production. Moreover, Brito et al. [6], reported that the attenuation of abiotic stress related to the use of kaolin causes the change of important physiological, morphological and biochemical mechanisms.

Disregarding water stress, olive leaves treated with kaolin had the highest level of total polyphenol detected on average (11,246.82 µg g$^{-1}$ dw) followed by leaves treated with seaweed mix (6950.79 µg g$^{-1}$ dw) and *Trichoderma* based product (6828.03 µg g$^{-1}$ dw) (Table 3). In general, without considering the water stress, all treatments caused an increase in polyphenolic compounds compared to the control with the exception pinolene, which showed the lowest level of polyphenols. Treatment with pinolene provoked an inhibitory effect on PAL activity, hence reducing the concentration of polyphenolic compounds. In literature a similar effect was linked to the antioxidant 5-hydroxybenzimidazole foliar application on "Koroneiki" olive trees [44].

The results of the antioxidant activity essays, carried out on the polyphenolics extracts of the olive leaves, were reported in Table 4 and expressed as TEAC (mmol Trolox kg$^{-1}$ dw). As shown, a significant interaction was present between the two tested factors.

**Table 4.** Antioxidant activity and total polyphenols content in leaves of olive tree grown in a greenhouse under optimal irrigation (100% irrigation) and limited water availability (50% irrigation) treated with several biostimulants: C = control, TR = Trichoderma, M = Micro-Algae, P = Seaweed mix, K = Kaolin, B = Glycine betaine, V = Pinolene.

| | DPPH | ABTS | FRAP | FOLIN |
|---|---|---|---|---|
| **100%** | | **(mmol trolox/kg)** | | **(mg/kg dw)** |
| C | 26.12 ± 0.44 defg | 87.59 ± 1.18 c | 95.74 ± 1.99 de | 2620.16 ± 33.57 fg |
| M | 27.47 ± 0.54 cde | 86.47 ± 2.21 c | 118.48 ± 0.23 c | 3744.19 ± 581.40 e |
| P | 29.17 ± 0.41 c | 105.33 ± 0.98 b | 138.01 ± 3.45 b | 9538.76 ± 637.77 b |
| TR | 28.76 ± 0.15 cd | 70.54 ± 4.09 de | 131.14 ± 2.23 bc | 8337.21 ± 58.14 c |
| B | 24.24 ± 0.44 gh | 62.87 ± 2.19 ef | 103.00 ± 5.16 d | 3608.53 ± 134.27 e |
| V | 26.32 ± 0.16 defg | 64.33 ± 7.53 ef | 96.84 ± 3.39 de | 2484.50 ± 67.13 fg |
| K | 36.43 ± 0.21 a | 130.59 ± 4.60 a | 166.62 ± 0.77 a | 18,375.97 ± 378.28 a |
| **50%** | | | | |
| C | 23.22 ± 0.96 h | 80.20 ± 2.44 cd | 79.42 ± 1.87 fg | 2426.36 ± 33.57 g |
| M | 24.77 ± 0.59 fgh | 80.11 ± 1.27 cd | 85.04 ± 1.05 efg | 2988.37 ± 209.63 f |
| P | 20.65 ± 0.52 i | 56.90 ± 2.42 f | 82.46 ± 1.56 efg | 2600.78 ± 33.57 fg |
| TR | 25.98 ± 0.70 efg | 74.83 ± 0.78 d | 83.70 ± 0.86 def | 5275.19 ± 412.48 d |
| B | 22.59 ± 0.47 hi | 62.61 ± 1.82 ef | 80.95 ± 3.14 g | 3511.63 ± 253.42 e |
| V | 27.01 ± 0.56 cdef | 78.82 ± 6.90 cd | 101.84 ± 2.22 d | 2620.16 ± 33.57 fg |
| K | 33.57 ± 0.44 b | 106.40 ± 1.04 b | 138.64 ± 29.63 b | 18,686.05 ± 100.70 a |
| W | ** | * | *** | ** |
| T | *** | *** | *** | *** |
| W × T | *** | *** | *** | *** |

Values are means ± standard error of three biological and three technical replicates. Asterisks indicate significant effect of limited water availability (W), biostimulant treatment (T), time of measurements (S) and their interaction (W × T × S) according to ANOVA (ns = not significant; * = $p < 0.05$; ** = $p < 0.01$; *** = $p < 0.001$). Different letters indicate significant differences based on Duncan's test ($p = 0.05$).

According to DPPH data, all treatments induced an increase in antioxidant activity compared to the untreated sample at 100% of evapotranspiration except for glycine betaine, for which a decrease in antioxidant activity was observed compared to the untreated sample. In particular, the antioxidant activity measured with DPPH assay was between 24.24 mmol kg$^{-1}$ (glycine betaine) to 36.43 (kaolin): leaves treated with kaolin showed the highest antioxidant activity, followed by pinolene, with both treatments being significantly higher than the control. In 50%, the values ranged from 20.65 mmol kg$^{-1}$ (seaweed mix) to 33.57 mmol kg$^{-1}$ (kaolin), whereas *Trichoderma*, pinolene and kaolin generated significantly higher DPPH compared to the control.

As for the ABTS test, the values obtained showed, at 100%, an improvement of antioxidant activity in correspondence with kaolin (+49.09%) and with seaweed mix treatment, (+20.25%), while a decrease in antioxidant activity of −19.47%, −28.22% and −26.56% was found, respectively, in the case of foliar application of *Trichoderma*, glycine betaine and pinolene. However, the treatment with micro-algae did not induce significant changes. For the water regime reduced by 50%, the values ranged from 56.90 mmol kg$^{-1}$ to 106.40 mmol kg$^{-1}$, and the lower value was referred to seaweed mix treatment. In fact, this last treatment caused a decrease in antioxidant activity compared to untreated leaves equal to −29.08%, while for treatment with glycine betaine the reduction observed was –21.93%. Once again, an increase in antioxidant activity was observed with the kaolin treatment (+32.64%), while insignificant changes were measured for micro-algae, *Trichoderma* and pinolene treatments.

In the case of FRAP, at 100%, the majority of the treatments exhibited an improvement in terms of antioxidant activity compared to untreated samples, expect for glycine betaine and pinolene. There was an average increase of 44.72% and kaolin treatment was the one that showed the highest increase (+74.03%). Similar results were recorded at 50% with

kaolin, which showed to be the most effective treatment leading to the highest increase (+74.57%) followed by pinolene (+28.23%).

Folin results were in line with those of mass spectrometry investigations and with those of the antioxidant activity evaluated with the FRAP method, showing both in well-irrigated samples and in that water stressed an overall improvement of the antioxidant performances of all plants treated with biostimulant compared to untreated control.

In general, water stress reduced both the antioxidant activity and the level of polyphenolic compounds. This decrease may be associated with the defence-related functions of polyphenolic compounds [45]. In literature it is reported that total phenolic and flavonoid contents in leaves of two olive cultivars (Gemlik and Kilis Yaglik) were significantly affected by irrigation treatments, with a cultivar dependent response [45]. In line with our results, Dias et al. [42] observed a reduction in the flavonoid pool after a water deficit of thirty days and attributed this development to the ROS scavenger capacity of flavonoids. Moreover, Dias et al. [42] reported that independently of the treatment, olive leaves are rich in the *o*-dihydroxy B-ring-substituted flavonoids such as luteolin-7-*O*-glucoside that could assist to this species' high-stress tolerance.

As regards biostimulant treatments, it is interesting to underline the remarkable effect of the kaolin that was responsible for both increased antioxidant activity and polyphenol content in both water regimes investigated compared to untreated control. This result could be due to the upregulation of the gene transcription encoding chalcone synthase and PAL, as antecedently stated for water stressed grape berries [43]. The results obtained showed that the different biostimulants differentially modulated olive leaves phenolic compounds content and antioxidant activity. Denaxa et al. [44] reported that kaolin engendered the highest total phenols concentration, whilst drought-stressed plants exhibited higher total flavonoids concentration when compared to the irrigated plants. However, in contrast to our data, the latter authors noted that glycinebetaine treated trees presented the highest oleuropein content just after foliar application, whereas kaolin treated trees presented the lowest one. In this same study it was featured that kaolin application engendered high activities of antioxidant enzymes such as glutathione reductase, peroxidase and superoxide dismutase under water deficit, compared to other alleviating products that were investigated. In our previous study we reported that *Trichoderma* strains may boost phenolic compounds concentrations, incrementing the plant nutrient uptake mechanism and meliorating plant nitrogen use efficiency, concomitantly with a positive influence on the antioxidant activity [19]. Similar effects were also reported, where the application of *Trichoderma harzianum* caused a significant increase in tomato fruit quality in terms of total soluble sugars, carotenoinds, antioxidant capacity, and polyphenolic content [46].

Pinolene, in agriculture, is used as film-forming antitranspirant that can prevent water loss from the arial part of a plant [47,48]. Several studies have highlighted the profitable effect of film-forming compounds, especially in horticultural crops [43–45]. Brillante et al. [8] reported that pinolene treatment on grape caused a decline in sugar content and anthocyanin level when compared to a control. Moreover, seaweed extracts are biostimulants traditionally used as soil conditioners as a scope to improve the growth of agricultural crops [49]. The effects of foliar application of algal extracts on the polyphenolic quali-quantitative profile of plants were reported in the literature [50–52] and showed the ability of these extracts to stimulate primary and secondary metabolism by improving nutrient uptake and assimilation, as well as favoring the synthesis and accumulation of phytochemicals which are important for human diet.

*3.4. Correlation between Total Phenolic Contents and Each Antioxidant Assays*

In Figure 2, the correlation between the total phenolic content (Folin) and each of the antioxidant capacity methods in both water regimes is shown.

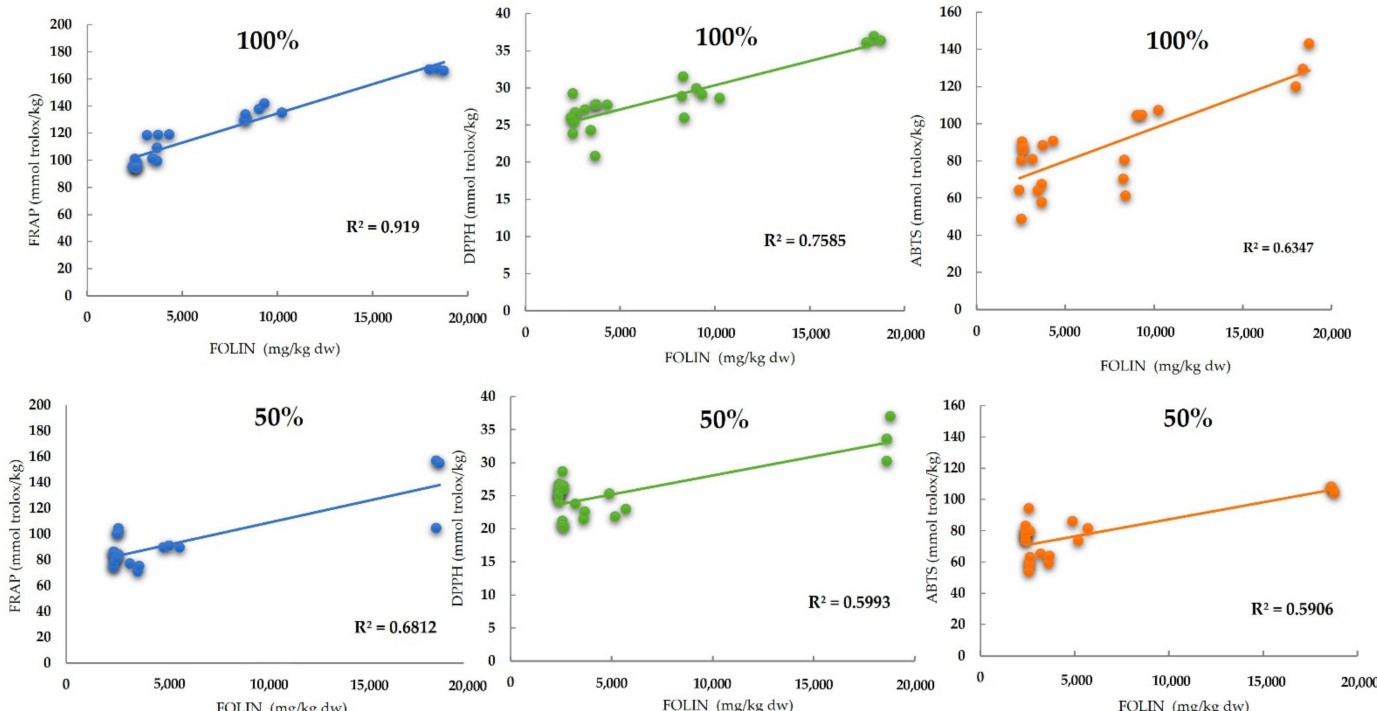

**Figure 2.** Correlation between the total phenolic content (Folin) and antioxidant capacities of leaves (FRAP, DPPH and ABTS assay) under optimal irrigation (100% irrigation) and limited water availability (50% irrigation).

In the 100% water regime there was a high and positive correlation between Folin and FRAP ($r^2$ = 0.919), Folin and DPPH ($r^2$ = 0.758), Folin and ABTS ($r^2$ = 0.635), while lower values of correlations were shown in 50% water regime with values of $r^2$ respectively equal to 0.681 (Folin-FRAP), 0.599 (Folin-DPPH) and 0.591 (Folin-ABTS). These results are in agreement with other studies, where a relationship was observed between the potential antioxidant activity, total phenolic and flavonoid levels of the extract in olive leaves [53]. There are some literature data revealing a strong correlation between the total number and content of phenolics and the antioxidant activity of food, medicinal, plants, fruits, or vegetables [54–56]. The weaker correlation between total polyphenol content and antioxidant activity, observed in 50% water regime could be attributed to the modification of quantitative polyphenolic pattern under stressful conditions. This can cause low correlations between different methods, taking into consideration that polyphenolic compounds have multiple activities and can scavenge radicals by different mechanisms [57].

### 3.5. Principal Component Analysis (PCA)

A principal component analysis (PCA) was done to feature the repercussion of the biostimulant treatments on the biometric, eco-physiological and qualitative parameters analyzed above. The first two principal components (PCs) disclosed 74.22% of the cumulative variance (Figure 3), with PC1 detailing for 41.86% and PC2 for 32.36%.

The PC1 was positively correlated with vegetative and nutraceutical parameters, while PC2 was positively correlated with eco-physiological parameters. PCA is effective in plotting the physiological, vegetative and nutraceutical parameters of the young olive trees in affiliation of the different biostimulant treatments and their utility. Kaolin treatment, in both water regimes 100 and 50%, was positioned in the downright quadrant of the PCA score plot, since it engendered the highest value of total polyphenols and antioxidant activity. In water regime 100%, all biostimulants were positioned in the upper right quadrant of PCA score plot showing positive correlations with RWC, stomatal conductance, leaves number, leaf area, and leaf dry matter.

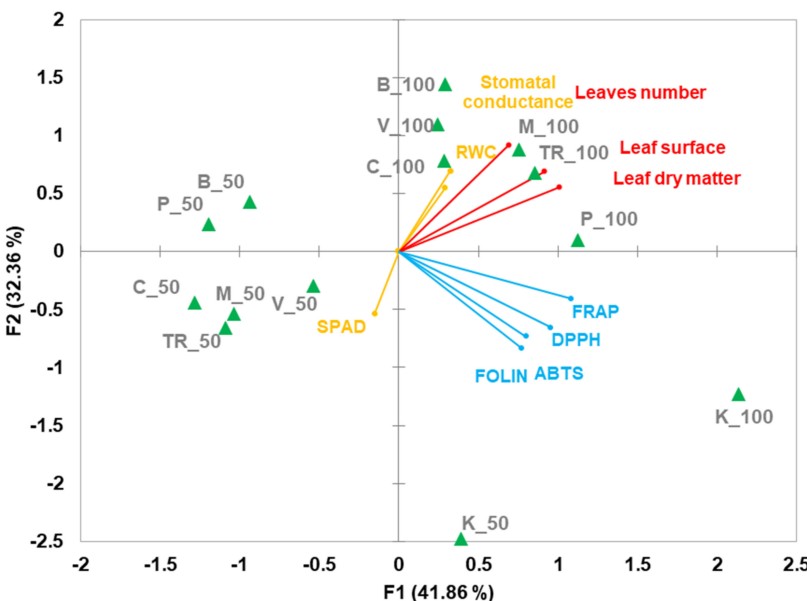

**Figure 3.** Principal component analysis (PCA) of vegetative parameters (leaf area, leaf dry weight and leaf number), eco-physiological parameters at the end of growing season (RWC, SPAD and stomatal conductance) and nutraceutical parameters (Folin, ABTS, DPPH and FRAP) in leaves of olive tree grown in a greenhouse under optimal irrigation (100% irrigation) and limited water availability (50% irrigation) treated with several biostimulants: C = control, TR = *Trichoderma*, M = Micro-Algae, P = Seaweed mix, K = Kaolin, B = Glycine betaine, V = Pinolene.

To conclude, we list in Table 5 a summary of the composition, the application procedure, application time and effects of the individual biostimulants that were used in this test on the leaves of young olive plants.

**Table 5.** Summary of the composition, the application procedure of the biostimulants and the effects on the olive leaves.

| Treatments | Composition | Application Procedure | Effects |
|---|---|---|---|
| *Trichoderma* (TR) | 1% *w/w Trichoderma harzianum*, strain T-22 spores ($1 \times 10^9$ spores/g) and 99% *w/w* inert ingredients | Drench application—6.67 g/L of water | Improves RWC values and total polyphenols content |
| *Micro-Algae* (M) | Free L-amino acids (4.1% *w/w*), total nitrogen (7% *w/w*), organic nitrogeno (5.6% *w/w*), nitric nitrogen (5.6% *w/w*), $P_2O_5$ (0.5% *w/w*), $K_2O$ (6.7% *w/w*) | Drench application—6.67 g/L of water | Improves SPAD values and stomatal conductance |
| *Seaweed mix* (P) | Organic carbon C (6%), mannitol 9 g/L | Drench application—4 ml/L of water | Improves SPAD and RWC values |
| *Glycine betaine* (B) | Glycine betaine, trehalose, plant extracts containing zeatin. | Drench application—10 g/L of water | Improves vegetative activity and eco-physiological parameters in the leaves |
| *Kaolin* (K) | Copper (Cu) totale 5% | Foliar application—40 g/L of water | Improves the polyphenol content and antioxidant activity in the leaves |
| *Pinolene* (V) | di-1-p-menthene (96%), coformulants, inert emulsifiers (4%) | Foliar application—10 mL/L of water | Improves vegetative activity and RWC values |

## 4. Conclusions

The results of this study highlight the importance of biostimulants' application to mitigate the effects of abiotic stresses (high temperatures and drought), with different effects

based on the product used. Regarding the vegetative parameters, significant differences were shown between the two watering regimes (100% and 50%), with higher values registered at 100%. Biostimulants' effects were evident in conditions of water stress, and glycine betaine treatment and algae products (micro-algae and seaweed mix) reported higher values in the increase in the number of leaves and leaf area; these same treatments showed positively significant values also regarding the eco-physiological parameters in both water regimes. Particularly interesting results were obtained with kaolin applications that caused a considerable two-fold increase in the total polyphenols content compared to the control, and a significant increase as well in the antioxidant activity. These results are interesting for improving the quality of olive oils characterized with low phenolic and antioxidant components, with the addition of leaves rich in polyphenols that can be used equally for pharmaceutical purposes. Future studies in the open field and on olive trees in full production will be necessary to evaluate the efficiency of biostimulants in mitigating damage from abiotic stress and to evaluate the effect on drupes and oil parameters.

**Supplementary Materials:** The following are available online at https://www.mdpi.com/article/10.3390/agriculture12020227/s1, Figure S1: Typical chromatograms observed for the extracts of olive leaves analyzed in this study and the mass specifications of the compounds of interest relative to phenolic compounds (separation via UHPLC). Table S1: Retention time and exact mass spectra data of apple polyphenols investigated by UHPLC-HRMS Orbitrap.

**Author Contributions:** Conceptualization, C.D.V., A.R. and Y.R.; methodology, C.D.V., C.E.-N. and G.G.; software, A.C. and C.E.-N.; validation, C.D.V., Y.R., A.R. and G.G.; formal analysis, A.C., P.G. and G.G.; investigation, A.C., C.E.-N., P.G. and G.G.; resources, C.D.V. and Y.R.; data curation, A.C., G.G. and C.E.-N.; writing—original draft preparation, A.C., G.G. and C.E.-N.; writing—review and editing, supervision, S.C., C.D.V., Y.R. and A.R.; project administration, Y.R. and C.D.V.; funding acquisition, Y.R. and C.D.V. All authors have read and agreed to the published version of the manuscript.

**Funding:** This research was funded by PSR Campania 2014/2020 Research Project Measure 10—Type of Intervention 10.2.1—Conservation of indigenous genetic resources to protect biodiversity (DICOVALE) —Vegetable Genetic Resources—CUP: B24I19000440009.

**Institutional Review Board Statement:** Not applicable.

**Informed Consent Statement:** Not applicable.

**Data Availability Statement:** Not applicable.

**Acknowledgments:** We would like to thank Koppert, Biogard, Manica, AgriAlgae®, Gobbi and Biolchim for providing the biostimulants products.

**Conflicts of Interest:** The authors declare no conflict of interest.

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
