# Peer review of "Biostimulants Improve Plant Growth and Bioactive Compounds of Young Olive Trees under Abiotic Stress Conditions"

_agriculture, doi:10.3390/agriculture12020227_

Round 1
Reviewer 1 Report
Major revisions are still needed:
1- Please provide detailed supp. Info with all fine research details.
2- Detailed statistical analyses should be descibed in research
3- Extensive English Editing by a native English speaker must be performed.
Author Response
Major revisions are still needed:
Dear Reviewer 1, extensive revisions were applied to this manuscript in order to make it adequate for publication on Agriculture MDPI and suitable up to the expectations of the journal readers.
1- Please provide detailed supp. Info with all fine research details.
Dear reviewer, the authors have added more details of the bioactive compounds analyzes in lines 154-239 and they have added some supplemental material, in particular a standard chromatogram of olive leaves polyphenolic extract (obtained using UHPLC-HRMS Orbitrap analysis) and the mass characteristics of the investigated polyphenolic compounds were added as supplementary data.
2- Detailed statistical analyses should be descibed in research
More information on statistical analysis has been added to the line 240-245: “the PCA results were shown as a biplot to highlight the interaction between samples and variables. Samples are displayed as points while variables are displayed as vectors.”
3- Extensive English Editing by a native English speaker must be performed.
An Extensive English Editing was done by a native English speaker in order to render the manuscript adequate for publishing.
Reviewer 2 Report
This experiment last for only 4 months only one season, in horticultural studies, especially with olive- it's a very short time. It's also not clear how much nutrients were available (excluding the treatments), you need to elaborate on the contribution of the fertilizer applied (Nitro- phoska Gold)
Title; please replace 'Nutraceutical' with a better term
Line 93: what is 'agricultural soil' which type of soil ?
Line 94-95, what is the composition of '“Nitro- phoska Gold”
Line 101, what is 'radical application ?
Line 104; , "a high quality biological biostimulant"- this is scientific paper not marketing leaflet, please use the right term
Line 119; please explain exactly how much water was applied and how you calculated it and also how exactly you irrigate and what was the frequency .
Line 118, if you irrigate every day in accordance to 50% of ET of the previous day, very soon you will barely irrigate and the plant will be dehydrated. Can you please explain how exactly you have done it?
Lines 125-127; you must add more information regarding the stomata conductance, how many measurements? how many measuring days ?, at what time of the day?
Line 138, how did you took the picture of the measured plants ?
Lines 185-191; how did you reached such high temperatures ? the plant were in greenhouse ? please add this information
Table 1 'leaf Dry Matter', please add units
Line 183; there is no different chapter for the 'Discussion' and it seems that you decided to combine the results with the discussion, so please write' Results and discussion' and change the numbering of the 'Conclusion from '5' to '4'.
Table 3 can be omitted from the body of the ms. if needed, you may add it as supplementary material
Author Response
This experiment last for only 4 months only one season, in horticultural studies, especially with olive- it's a very short time.
Dear reviewer, thank you for your consideration. The aim of this study was to test six different biostimulants in order to select the best performing 3. These latter are to be tested in open field trial (that is currently in progress and it will last for 2 years as adopted for arboriculture experiments). We precised in the abstract and the conclusions that the experiment should be repeated in open field conditions on mature olive trees in full production.
It's also not clear how much nutrients were available (excluding the treatments), you need to elaborate on the contribution of the fertilizer applied (Nitro- phoska Gold).
Nitrophoska Gold is a slow-release fertilizer based on N (15%), P2O5 (9%), K2O (15%) and micronutrients ( Fe, B,Cu, Zn, Mn) . This information has been added at line 91. Anyhow this fertilizer was applied equally for all the plants belonging to the different treatments including the control.
Title; please replace 'Nutraceutical' with a better term
As suggested by the reviewer, the authors changed the term “Nutraceutical” to “Bioactive Compounds”
Line 93: what is 'agricultural soil' which type of soil?
Dear reviewer the soil used was clayey soil, the authors changed the term 'agricultural soil’ to a more suitable one “clayey soil” (Line 89).
Line 94-95, what is the composition of '“Nitro-phoska Gold”
Dear reviewer, the authors, as previously mentioned, have added the Nitrophoska gold composition to line 91. It is a slow-release fertilizer based on nitrogen, phosphorus and potassium.
Line 101, what is 'radical application?
Dear Reviewer, ‘Radical application’ was substituted by “was applied to the root system by irrigation” throughout the manuscript. Whereas in table 5, it was substituted by drench application.
Line 104; , "a high quality biological biostimulant"- this is scientific paper not marketing leaflet, please use the right term
The authors, as suggested by the reviewer, chose to delete the term “a high quality”, line 141
Line 119; please explain exactly how much water was applied and how you calculated it and also how exactly you irrigate and what was the frequency .
Dear reviewer, thank you for pointing out these important details. Therefore the following was elaborated.
The authors calculated the evapotranspiration of the well watered plants (100% watering regime group) using a gravimetric method as follows: every two days the pots were weighed and water loss by evapotranspiration was calculated. Subsequently, 100% or 50% of water loss corresponding to x mL or x/2 mL water per pot, respectively, was restored by drip irrigation. The drip irrigation system was con-trolled by a programmable timer and it was powered by an electric pump, feeding water to drippers at a flow rate of 2L/h. Two or one drippers were installed into each pot for irrigation at 100% or 50%, respectively.This information has been added at lines 116-124.
Line 118, if you irrigate every day in accordance to 50% of ET of the previous day, very soon you will barely irrigate and the plant will be dehydrated. Can you please explain how exactly you have done it?
Dear reviewer, the authors specified that even at 50% the ETP was always calculated in reference to 100% (well irrigated plants) and then the value was halved. We hope that the new description mentioned above made the concept clearer. We remain available for any clarification.
Lines 125-127; you must add more information regarding the stomata conductance, how many measurements? how many measuring days? at what time of the day?
The authors added more details about stomatal conductance measurements as requested by the reviewer.
Stomatal conductance was determined at 12:00 a.m. (it has been added at line 131). All the eco-physiological measurements (including stomatal conductance) were carried out at the start and at the end of the growing season, taking 6 measurements per treatment. More precisely, the first measurement was made in June (one month after the first biostimulants application) and in September (at the end of the growing season). This information is present at the lines 145-158.
Line 138, how did you took the picture of the measured plants?
Dear reviewer, as reported in the text in line 140, the image for the leaf surface was acquired and processed with imageJ software version 1.50.
Lines 185-191; how did you reached such high temperatures? the plant were in greenhouse ? please add this information
Dear reviewer, as already reported in the text at the line 86 of M&M section, the test was carried out in a greenhouse where higher temperatures are expected in summer season in Mediterranean climate.
Table 1 'leaf Dry Matter', please add units
Thanks for this correction. The authors added the unit to Table 1.
Line 183; there is no different chapter for the 'Discussion' and it seems that you decided to combine the results with the discussion, so please write' Results and discussion' and change the numbering of the 'Conclusion from '5' to '4'.
Sorry for the inconvenience. The authors changed the paragraph number (line 135) and they added "results and discussion" (line 189).
Table 3 can be omitted from the body of the ms. if needed, you may add it as supplementary material
Table 3 was added to the supplementary material (Table S1).
Reviewer 3 Report
Overview and general recommendation:
Thanks for the opportunity to review this research. The manuscript entitled „Biostimulants Improve Plant Growth and Nutraceutical Parameters of Young Olive Trees Under Abiotic Stress Conditions” have described the efficiency of biostimulants in mitigating the damage from abiotic stress on young olive trees, by improving some vegetative, eco-physiological and leaf nutraceutical parameters. The subject of the manuscript is topical.
- Тhe title is clear and precise, as is the abstract;
- The introduction is clear;
- The used methods are accurate;
- Тhe figures and tables are well described;
- The results are well presented and described.
There are some technical errors in the text and the references. Please check them. The length, quality and language of the paper are adequate.
Author Response
Overview and general recommendation:
Thanks for the opportunity to review this research. The manuscript entitled „Biostimulants Improve Plant Growth and Nutraceutical Parameters of Young Olive Trees Under Abiotic Stress Conditions” have described the efficiency of biostimulants in mitigating the damage from abiotic stress on young olive trees, by improving some vegetative, eco-physiological and leaf nutraceutical parameters. The subject of the manuscript is topical.
- Тhe title is clear and precise, as is the abstract;
- The introduction is clear;
- The used methods are accurate;
- Тhe figures and tables are well described;
- The results are well presented and described.
There are some technical errors in the text and the references. Please check them. The length, quality and language of the paper are adequate.
Dear reviewer, thank you for your comment. The authors have modified some technical errors in the text and references. In particular:
Throughout the text the abbreviation TS (control thesis) has been changed to C, and the abbreviation C to K (kaolino);
Line 289: the paragraph name in Results and Discussion has been updated;
Line 335: the unit of measure has been added of the Leaf dry Matter;
Line 446: the term "hydroxycinnamic acids" has been updated;
Line 135: the paragraph has been numbered correctly;
Line 284: the reference has been updated according to the format of the journal.
Round 2
Reviewer 1 Report
Manuscript csn be accepted